# A novel class of polymeric fluorescent dyes assembled using a DNA synthesizer

Tracy Matray[1]*, Sharat Singh[2], Hesham Sherif[1], Kenneth Farber[1], Erin Kwang[1], Michael VanBrunt[1], Eriko Matsui[3], Hiroaki Yada[3]

1 R and D Department, Sony Biotechnology, Bothell, Washington, United States of America, 2 Independent Consultant, Sony Biotechnology, Bothell, Washington, United States of America, 3 Medical Business Group, Sony Imaging Products & Solutions Inc., Atsugi-shi, Kanagawa, Japan

* Tracy.Matray@sony.com

**Data Availability Statement:** All relevant data are within the paper and its Supporting information files.

**Funding:** All work funded by Sony Corporation and Sony Imaging Products & Solutions Inc. The

## Abstract

In the pursuit of a novel class of fluorescent dyes we have developed a programmable polymer system that enables the rational design and control of macromolecular constructs through simple control of polymer primary sequence. These polymers are assembled using standard phosphoramidite chemistry on a DNA synthesizer which allows for extremely rapid prototyping and enables many permutations due to the large selection of phosphoramidite monomers presently available on the market. This programmability to some extent allows us to control the interactions/spacing of payload molecules distributed along the designed polymeric backbone. Control of molecular architecture using this technology has allowed us to address the long-standing technical issue of contact quenching between fluorescent dyes offering new possibilities in the life sciences arena. Much like peptidic sequences coding for enzymes, cofactors, and receptors (all needing control of tertiary structure for proper function via primary sequence) our programmable system approaches a similar endpoint using a phosphate based polymeric backbone assembled in a completely automated fashion. Using this novel technology, we have efficiently synthesized several types of fluorescent dyes and demonstrated the programmability in molecule design, including the increases in brightness of the fluorescence emission.

## Introduction

Fluorescent dyes are ubiquitous in modern day science due to their intrinsic ability to function as a reporter group enabling easy and efficient detection of target species in an extremely sensitive manner. Creative use of fluorescent molecules has promoted rapid growth in a multitude of areas including sequencing, imaging, PCR, flow cytometry, proteomics and FISH. While these unique molecules continue to play a critical role in scientific research, relatively few new classes have been developed beyond the classical small organic molecules such as xanthates, coumarins, cyanines and the protein type dyes like Phycoerythrin (PE). Additional fluorescent molecules with high absorptivity and emission characteristics that are excitable using currently available lasers are needed to improve ultrasensitive multiplexed biomarker detection. Two

funder provided support in the form of salaries for all authors, but did not have any additional role in the study design, data collection and analysis, decision to publish, or preparation of the manuscript. The specific roles of these authors are articulated in the 'author contributions' section.

**Competing interests:** The commercial affiliation does not alter our adherence to PLOS ONE policies on sharing data and materials.

notable types that have proven useful include the polymeric light harvesting system developed by Sirigen and Quantum dots (Q-dots) [1–3]. Both classes have augmented the dye field to some extent but there remains opportunity to further enhance performance in both FCM and other application areas. In pursuit of a novel class of fluorescent dye we have initiated a program to investigate the development of unique dyes for use on both classical and newer spectral variants of cell analysis flow cytometry instruments [4].

Our initial investigations were inspired by work reported from the Kool group on a novel oligomeric dye system which used a ribose phosphate DNA backbone as a scaffold to control dye architecture [5,6]. This oligodeoxyfluoroside (ODF) system replaced standard DNA nucleobases with small fluorescent chromophores such as pyrene, perylene and stilbene. These short oligomers were assembled using standard phosphoramidite coupling chemistry (Fig 1) on a DNA synthesizer. The polymerization cycle begins with acidic deprotection of the dimethoxytrityl group from the 5'-hydroxyl residue of the solid support bound terminal nucleoside. This hydroxyl group is then coupled (in the presence of a suitable activator, usually tetrazole) to the next programmed phosphoramidite residue using the phosphine group on the 3'-hydroxyl group of the added monomer. Any unreacted support bound chains are then capped (to prevent truncation sequences from extending) using acetic anhydride. The last step involves treatment with iodine to oxidize the newly formed phosphine into the more stable phosphate linkage. This coupling chemistry is the current industry standard and is depicted in Fig 1 [7].

The ribose-based DNA backbone of ODFs promote stacking of the proximal fluorophores just like the natural nucleobases are preorganized in single stranded DNA, but to an even greater extent in order to minimize surface area and effectively reduce hydrophobic interactions with the aqueous environment. Upon excitation with the appropriate light source the stacked fluors in the ODF system created unique excited state species such as excimers and exciplexes [8,9]. These dyes were sequence specific in that different colors could be coded for by simple changes in the primary sequence of the monomeric dyes. We were intrigued by the potentially limitless number of permutations arising from this approach.

During our preliminary development of DNA based polymeric dyes we chose to focus on brightness enhancement/control as an initial goal. Many dye classes have been successfully

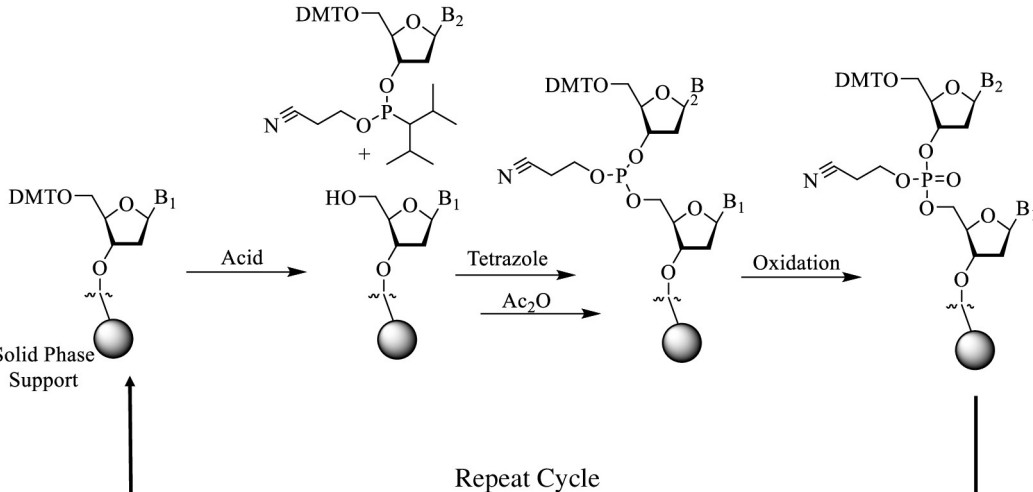

**Fig 1. Reaction scheme showing steps of standard DNA synthesis cycles.** Specific conditions used for the compounds reported below are provided in the materials and methods.

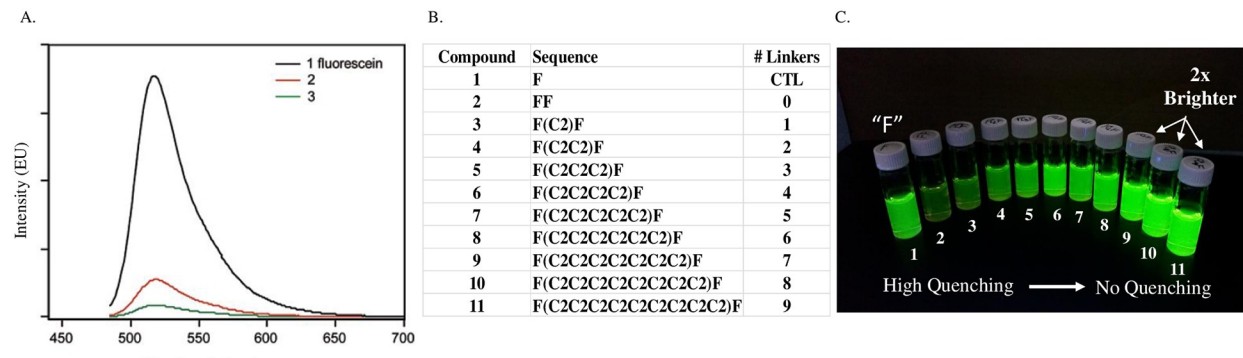

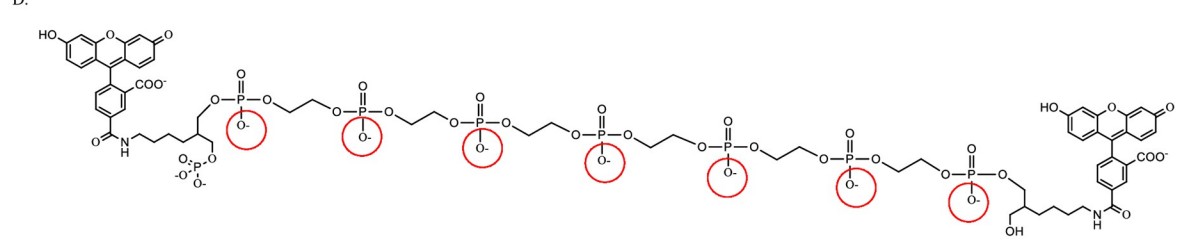

**Fig 2. Initial data testing the concept of using charge density to control polymer structure and prevent quenching.** A) Literature example of emission comparison of DNA targeting sequences containing one, two, or three fluorescein chromophores. A single reporting fluorescein unit emits substantially more than similar sequences with either two (red line) or three (green line) reporters. (Reprinted with permission from Eric T. Kool.) B) List of proposed polymeric dye sequences to test theory of using charge repulsion to spatially separate dyes (in this case fluorescein) and control quenching phenomena. C) Equimolar solution of constructs 1–11 from panel B (1 on far left through 11 on far right). See Materials and methods for solution preparation and emission measurements. D) Representative chemical structure of compounds used for testing (compound **8** containing six C2 spacers) with hard charges shown in red.

conjugated to DNA via direct incorporation into the backbone using standard phosphoramidite coupling chemistry or via post synthetic conjugation using NHS ester forms of the dye [10–12]. As such one might think that simply adding more dye monomers to the DNA sequence is all that would be required to create a brighter species. This has been reported not to be the case from multiple sources due to a phenomenon called contact quenching (Fig 2A) [6]. If multiple dyes are in proximity to one another, upon excitation they will preferably undergo contact quenching rather than the desired path of fluorescence via photon emission. Our idea to solve this problem focused on the manipulation of charge density to control the architecture and distance of dye spacing along a DNA based oligomeric backbone.

## Results and discussion

We believed that the highly charged nature of DNA (due to its phosphate backbone) afforded an opportunity to potentially control three-dimensional macromolecular structure; specifically, our aim was to use charge repulsion to spatially orient dye molecules along a polymer backbone. Automated DNA synthesis provided the perfect platform to manipulate charge density in a completely predetermined manner as the result of a phosphate moiety (hard negative charge) being formed during each monomer addition to the growing polymer chain (Fig 1). (The negative charged is liberated during ammonia deprotection by removal of the cyanoethyl group). It was hoped that maximizing charge density along the backbone would rigidify (through charge repulsion) and thus allow for separation of two fluorophores attached to

**Fig 3. Monomer phosphoramidite building units used to make polymer dye constructs reported herein (as named in catalogue).** All are commercially available (see Materials and methods): **12:** DMT-Ethane-Diol Phosphoramidite (C2); **13:** DMT-Propane-Diol Phosphoramidite (C3); **14:** DMT-Butane-Diol Phosphoramidite (C4); **15:** DMT-Hexane-Diol Phosphoramidite (C6); **16:** DMT-Tetraethyloxy-Glycol Phosphoramidite (Spacer 12—TEG); **17:** DMT-Hexaethyloxy-Glycol Phosphoramidite (Spacer 18—HEG); **18:** DMT-Polyethyleneglycol 1000 Phosphoramidite; **19:** DMT-6-FAM Phosphoramidite; **20:** Fmoc-Amino-DMT C7 CED phosphoramidite.

either end of a "dumbbell" sequence. To test this theory, a series of simple constructs was designed (Fig 2B and 2D (**compounds 1–11**)) which could be rapidly accessed using an automated DNA synthesizer. These constructs were designed to incrementally and systematically increase both charge density and distance between two fluorescein chromophores using commercially available phosphoramidite linker monomers (Fig 3). For the initial studies the smallest possible linker monomer (C2 spacer: compound **12**) was chosen to enable the most precise spacing adjustments possible. The control construct (designed for maximum quenching **compound 2**) has fluorescein coupled back to back with no linker units while a single fluorescein coupling (**compound 1**) represents maximum fluorescence possible for one unit of fluorochrome.

All compounds were made on a 1–2 μmol scale using standard DNA synthesis coupling conditions (see Materials and methods and Fig 1). After standard aminolysis deprotection RP-HPLC analysis showed high crude purities for most of the constructs (>**80%**) indicating reasonable coupling efficiencies. A table showing all theoretical and determined masses for these oligomers is provided in the supplementary information (S1 Fig). Compounds **1–11** were used without further purification for spectral analysis. Equimolar (25 nM) solutions of **compounds 1–11** were made using 100mM sodium bicarbonate (NaHCO_3) buffer at pH 9.5 and compared visually while being irradiated using a wide bandwidth photo illuminator (Fig 2 **panel C**). Visual comparison of the solutions of **compounds 2–9** from left to right clearly shows a sequential enhancement of brightness as more linkers (charge and distance) are inserted between the two fluorescein moieties. This represents a controlled removal of quenching as the fluorophores are separated at increasing distance along the backbone. The solution displays a full 2-fold brightness increase (compared to **compound 1**) for **compound 9** which correlates to seven C2 spacing units between the fluorophores. **Compounds 10 and 11** show

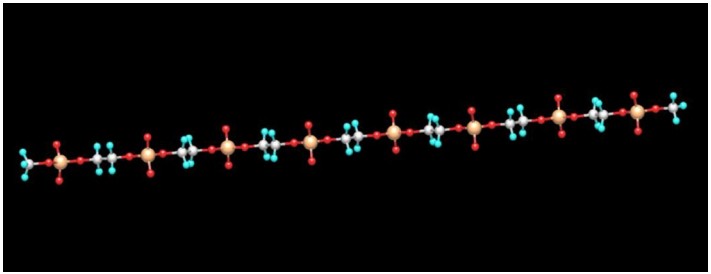

**Fig 4. MOPAC calculation showing minimized lowest energy conformation of polyphosphate Gen I dye.**

no further increases in brightness indicating that quenching effects have been completely removed and maximal brightness achieved for a two-fluorophore dumbbell construct.

To provide further support that the backbone structure can act as a rigid linker (due to charge repulsion) separating the fluorophores, a modeling study was undertaken to ascertain the lowest energy conformation of the experimental backbone structure. Fig 4 represents the results of this calculation. Details of the parameters used for this calculation are provided in the Materials and Methods section. As clearly depicted the lowest energy conformation adopts a linear chain.

Taking full advantage of the automated nature of DNA synthesis, sequences were then prepared by multimerizing the optimized dumbbell (**compound 9**) into sequentially longer and longer polymer chains comprised of multiple repeating cassettes (Fig 5A). All sequences were prepared in a similar manner as described above (Fig 1) and mass spectral analytical data is provided for each in the supplementary information (S2 Fig). The expectation was that each additional cassette added (containing a single fluorescein unit and seven $C_2$ spacer units (other spacers described below)) would result in a linear (and predictable) increase in brightness to the growing polymer (Fig 5A). Upon spectral analysis of equimolar solutions (10 nM) of these polymer constructs the brightness was shown to uniformly increase each time a repeat segment of $(C_2)_7F$ was added Fig 5B. Polymer **compound 25** was programmed to be 10x brighter (9 additional fluorophores) and spectral analysis showed exactly a 10-fold increase in

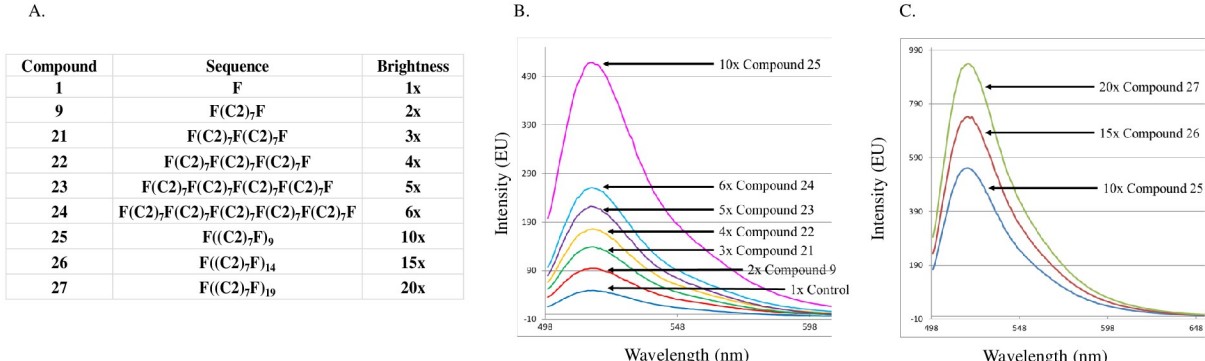

**Fig 5. Emission data showing the direct increase of brightness by increasing oligomer/polymer length.** A) Table of compounds prepared via DNA synthesis cycles (see Materials and methods) to investigate the programmable brightness aspect of phosphate-bridged polymers (F = Fluorescein, $C_2$ = **compound 12**). B) Emission curves for equimolar (10nM) solutions of **compounds 1,9, 21–25** (excitation at 488 nm; 100mM sodium bicarbonate ($NaHCO_3$) buffer at pH 9.5). C) Emission curves for equimolar (10nM) solutions of **compounds 25–27** (excitation at 488 nm; 100mM sodium bicarbonate ($NaHCO_3$) buffer at pH 9.5).

brightness compared to the control solution of fluorescein alone. Even longer programmed sequences of 15x and 20x were prepared (**compounds 26 and 27**) and these too show linear increases in brightness (Fig 5C).

It should be noted that **compound 27** required 153 couplings on the DNA synthesizer to reach full length. As mentioned, before we attribute the ability to access extra-long sequences of this type (in reasonable purity and yield on small scale) to the nature of the monomeric phosphoramidite building blocks (**12 and 19**). All reactive alcohols are primary in nature and the overall size of the building blocks is quite small. This allows for greater coupling efficiencies than are normally observed for standard DNA synthesis with the natural bases (A,C,G and T).

To further explore the possible physiochemical effects of different linker compositions, additional constructs were designed and synthesized using alternative spacing units (Fig 3, **compounds 16–18**). A series of 36 different dumbbell constructs was made for testing, each comprising of two fluorescein molecules separated by increasing numbers of each linker. Each data point in Fig 6 shows the brightness level (degree of quenching) of each of the 36 constructs in 6 separate sets of data (one for each linker type). The expectation was that in general as the linker length increased the overall number of spacers needed to fully prevent quenching would decrease. The results of the linker study (Fig 6) verify that indeed in general as the length of spacer unit increases the overall number of units required to prevent all quenching decreases. The curves shown in Fig 6 shift from right to left in sequential order going from the shortest spacer containing two carbon atoms (C2 **compound 12** grey data) to the longest spacer HEG (**compound 17** orange data) which equates to 18 atoms per spacer unit. More specifically where the C2 linker requires seven units to prevent quenching C3 only needs six, and the C4/C5 spacers need five units if one assumes an EU reading of 500 to be near the targeted doubling of brightness for these types of dumbbell molecules Fig 6.

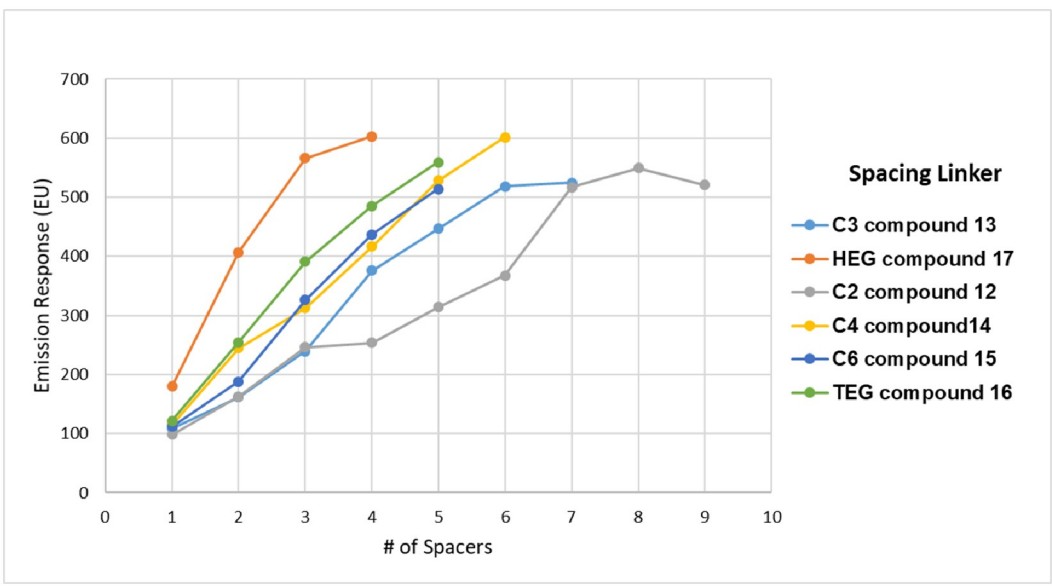

**Fig 6. Overlay plots of brightness versus number of linker units for a series of six different alternative linkers (Fig 3) used to make dumbbell type compounds containing two end unit fluorescein molecules.** All solutions were measured at equal concentrations (excitation at 488 nm; 100mM sodium bicarbonate (NaHCO$_3$) buffer at pH 9.5). Using C2 as a control 500 EU (emission units) correlates to doubling of brightness and full removal of quenching.

The shorter ethylene glycol-based linkers (**compounds 16 and 17**) clearly need the fewest units (approximately 3–4) to achieve removal of quenching as indicated by their representative curves being furthest to the left in Fig 6. This has obvious implications for manufacturing costs but more importantly it also allows for more precise control of the charge density of these molecules. Generation I polyphosphate dyes (use of linker C2 (**compound 12**)) have very high charge density (8 phosphates in each repeat cassette (dye plus seven C2 linkers = cassette) and are rigid (Figs 2D and 4). Generation II dyes (containing ethylene glycol type spacers **16 and 17**) have less charge (4 phosphates (3 HEGs)) associated with the backbone but are still capable of controlling dye quenching Fig 6. It is also interesting to note the role of charge by comparing the GEN II dumbbell molecule with two linkers to the GEN I molecule (C2 spacers) with seven linkers. Each combined spacer unit has almost identical number of atoms in the chain (length) but the GEN II construct is roughly 20% less bright. We attribute this difference to enhanced rigidity imparted by the addition charge density. Additional direct evidence was obtained when both polymer types were treated with increasing concentrations of magnesium chloride. One would predict that if negative charge were playing a role, the addition of salt would diminish overall brightness. This is exactly what was observed experimentally (S3 Fig).

A similar linearity study was performed for longer repeat sequences of the optimal GEN II cassette (3 HEG spacer with one dye), and it was shown that this polymer type dye behaves in a programmable nature as well in terms of a direct relationship between polymer length and brightness (S4 Fig). Synthesis of GEN II type polymers also showed reasonable coupling efficiencies using identical phosphoramidite coupling as depicted above (Fig 1) (>98.5%). Some were made on larger scale (13 µmol) using an AKTA 100 synthesizer (S5 Fig). This suggests (not yet proven) that these types of oligomers/polymers could be produced at even larger scale as the pharmaceutical industry routinely makes DNA sequences on kilogram scale for species like antisense drugs. Performing a modeling study on the optimized GEN II based backbone (3 HEG spacers) revealed an elongated coil-based structure for the polymer Fig 7. The hope was that the more compact structure of GEN II type polymers in conjunction with lower charge densities would result in better performance when used in flow cytometry-based applications.

Next an attempt was made to reduce charge even further to help guard against potential repulsive interactions from the dyes with cell surfaces (lower Kd of antibody). With this in mind PEG phosphoramidite linker (**compound 18**) was used to make a series of constructs to determine its suitability as spacer for these types of programmable species. It was discovered

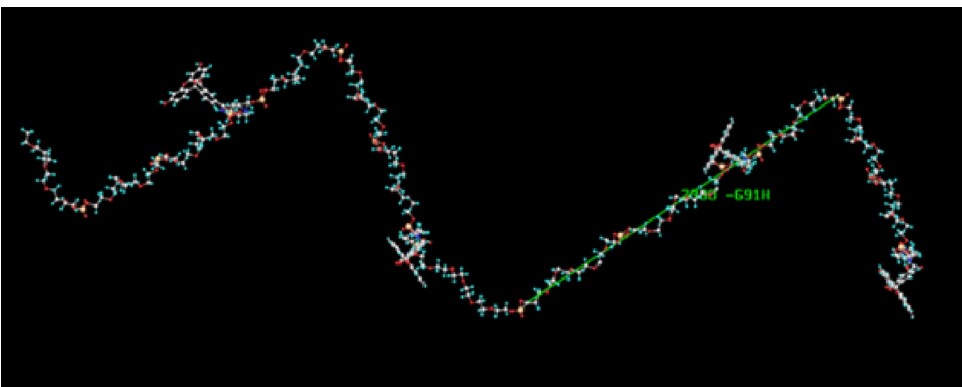

**Fig 7. MOPAC calculation showing minimized lowest energy conformation of polyphosphate Gen II dye.**

A.                                    B.

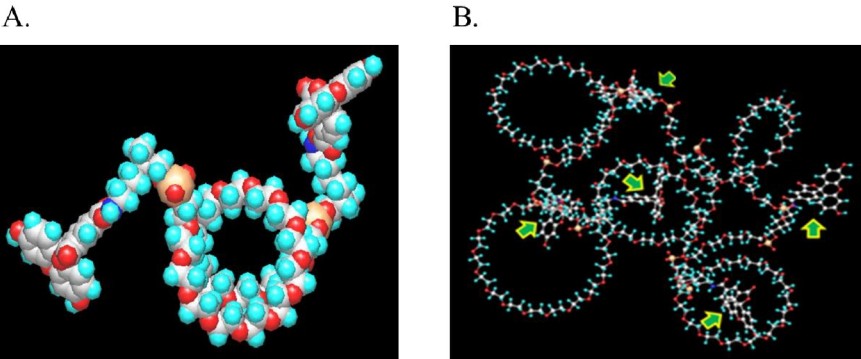

**Fig 8. MOPAC calculation showing minimized (lowest energy conformation of 1000kd PEG Gen III dye.** Green arrows indicate dye chromophores.

that only a single unit of this linker was required to lower the quenching between two fluoresceins in a dumbbell construct. However, it was discovered that unlike the shorter HEG linker and methylene based C2 spacer, the PEG based backbone was unable to maintain full linear increases in brightness beyond 5 cassettes (some suppression observed) (S6 Fig). Interestingly when a modeling study was performed on this type of backbone a more compact coil system was predicted (Fig 8). A possible explanation for this is that for shorter sequences of GEN III (contains PEG linker **18**) compounds, hydrogen bonding along the PEG units is the main force determining the macromolecular structure, not charge repulsion. Bridging with sodium ions could also play a role in these coil forms.

All the dye polymers presented thus far have used fluorescein as the dye component. This choice was mainly for expediency of synthesis as well as cost savings purposes (fluorescein being one of the least expensive phosphoramidite dyes). Unfortunately, the commercial availability of fluorescent dyes as extendable phosphoramidite monomers is somewhat limited. This is primarily due to complex protecting group strategy required for synthesis in most cases, as well as many of the dyes not being stable to standard basic deprotection conditions. Basic treatment of some type is typically required for deprotection (removal of β-cyanoethyl groups) and cleavage of the polymers from the solid phase support after automated synthesis. For this novel dye class to be broadly applicable it would require adaptation to enable other dye types to be adorned on the core backbone constructs described previously. It was envisioned that if a suitable attachment point could be introduced into the core backbone wherever a dye entity was desired, then the dye could possibly be added post synthetically. This is common practice for the labeling of oligonucleotide sequences with dyes that are not available as phosphoramidites (examples: Texas Red, Aquaphlor Dyes, AlexaFluors and Bodipy) [12]. Almost all commercial dyes of relevance are available as NHS (N-hydroxy succinimide) activated forms for conjugation to antibodies via covalent bonding to amine groups. As such we chose to incorporate amino functional groups into the core polymeric species using the f-moc amine protected linker phosphoramdite **20** (Fig 3).

A generic structural representation of a preprogrammed polyamine sequence is shown in Fig 9 with five amino attachment points depicted and X representing one of the three linkers types (**compounds 12,17,18**). In theory any number of such attachment points can be programmed into the backbone with any desired spacing due to the automated nature of the synthesis. It should also be emphasized that not just dye payloads can be loaded onto these types of polymers, in fact any activated small molecule can potentially be loaded onto the polyamine

**Fig 9. Generic programmable polyamine where X is one of the three linker types used in this report (compds 12,17,18).**

depicted in Fig 9. Interestingly, this approach failed during initial attempts to load activated NHS dyes onto polyamino sequences constructed using the GEN I $C_2$ linker (**compound 12**) in the proper configuration to prevent quenching (seven repeat units (Fig 2D). Analysis of reaction products revealed only one or two successful dye additions to the polyamine backbone. Even using a large excess of equivalents of NHS activated dyes for the loading reaction resulted in poor addition to the backbone. A possible explanation for this observation is that the numerous phosphate residues in these $C_2$ based polymers (GEN I) were acting in a self-buffering mechanism to protonate the amino residues and thus prevent successful coupling of the activated dye residues.

A different result was obtained however, when X was changed to the HEG (**compound 17**) or PEG (**compound 18**) linker type. Complete loading of various activated dye types was achieved (as indicated by LCMS analysis S7 Fig) for various lengths of polymers containing increasing numbers of attachment points (3, 5, and 10 amines). Upon isolation equimolar solutions of these constructs were prepared and used to measure relative brightness levels using a standard fluorimeter. (Fig 10). As clearly indicated all tested dyes showed similar patterns of brightness enhancement as more attachment points were added to the polymer. Reducing the phosphate density appears to have mitigated the undesirable self-buffering observed for the higher phosphate density backbone of the $C_2$ system. It should be noted that an additional orthogonal attachment chemistry has been developed which enables up to three different types of dyes to be deployed along these types of polymers at predetermined positions (Manuscript in preparation).

## Materials and methods

### Polymeric dye synthesis

Commercially available phosphoramidites including spacers (HEG, PEG, and various length carbon chains such as C2, C3, C4, etc.) and fluorescein dye were purchased from Glen Research (Sterling, VA) or ChemGenes Corporation (Wilmington, MA). The solid support used was either a long-chain alkylamine controlled pore glass (LCAA-CPG) with 1000 Angstrom pores obtained from Link Technologies (now LGC Genomics, Teddington, UK) or a polystyrene resin from GE Healthcare (Marlborough, MA). The supports used were all loaded with either a DNA base or phosphate at loadings from 30–300μmol/g. Anhydrous DNA synthesis reagents were all purchased from either Glen Research or EMD. All oligonucleotide dyes were synthesized on an ABI 394 DNA/RNA synthesizer using standard protocols for the phosphoramidite based coupling approach. Phosphoramidite monomers were typically

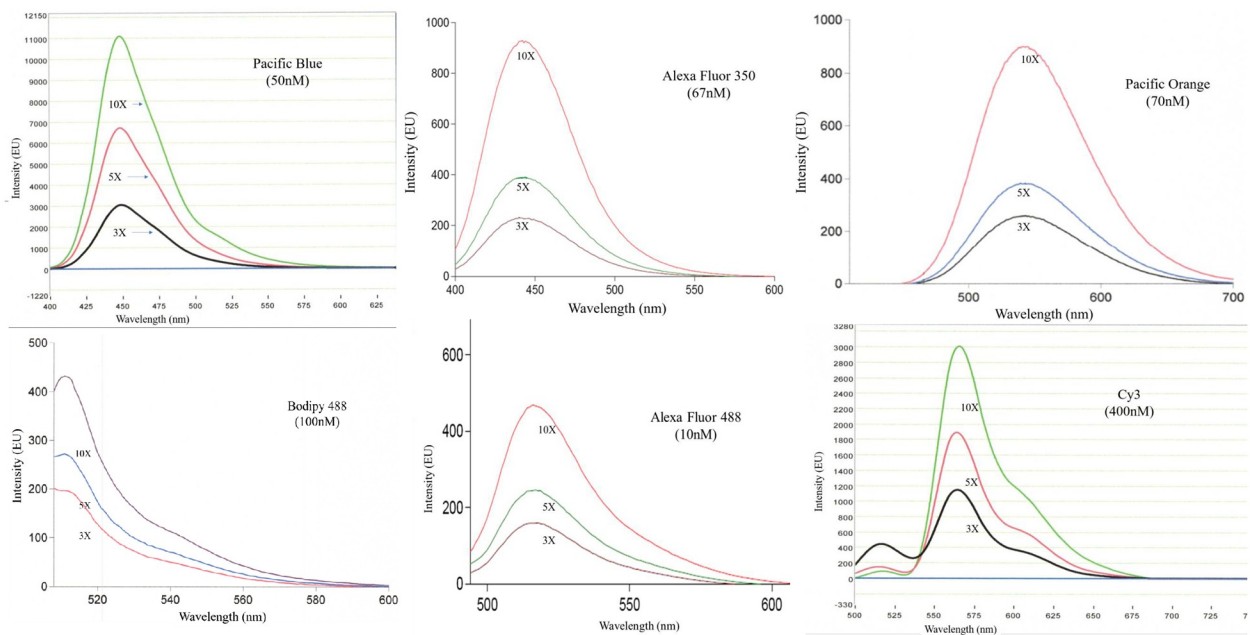

**Fig 10. Emission spectra for various fluorophores attached to programmable polyamines.** Samples of each dye on 3x, 5x, or 10x polyamine backbones were standardized to equimolar concentration (in parenthesis) and analyzed on a ThermoFisher Nanodrop 3300 fluorimeter (Pacific Blue and Cy3) or Cary Fluorimeter (Bodipy 488, Alexa Fluor 350 and 488, Pacific Orange): A. Pacific Blue (50nM) B. AF350 (67nM) C. Pac Orange (70nM) D. Bodipy 488 (100nM) E. AF488 (10nM) F. Cy3 (400nM). Polymer-fluorophore constructs showed increasing fluorescence intensity with an increase in the number of fluorophores attached to the polymer.

dissolved in 100% anhydrous acetonitrile but some dyes would require other solvent mixtures for optimal solubility (10–50% addition of tetrahydrofuran, dichloromethane, etc.). The chain assembly cycle for the synthesis of the dye constructs was as follows: (i) detritylation, 3% dichloroacetic acid in dichloromethane; (ii) coupling, 0.1 M phosphoramidite and 0.25 M 5-(ethylthio)-1H-tetrazole in acetonitrile; (iii) capping, acetic anhydride in THF/lutidine, 1/1, v/v; (iv) oxidation, 0.02 M iodine in pyridine/water, 9/1, v/v. Chemical steps within the cycle were followed by acetonitrile washing and flushing with dry argon. Cleavage from the support and removal of base and phosphoramidite protecting groups was typically achieved by treatment with concentrated ammonium hydroxide for 2 hours at 55 degrees Celsius. For some dye constructs milder conditions (potassium carbonate, no heat, shorter incubation, etc.) were used depending on the dye attached. Oligonucleotide dyes were then analyzed by reverse phase HPLC.

## Reverse phase HPLC analysis

The system used is a Waters Acquity UHPLC with a 2.1mm x 50mm Acquity BEH-C18 column held at 45˚C. Method is 1% B held for 3 minutes then ramped to 60% B over 10 minutes (where B is methanol and A is 100 mM 1,1,1,3,3,3-hexafluoro- 2-propanol [HFIP] in 8.6 mM triethylamine, pH 9). Molecular weight identity was determined with a Micromass Quattro Micro MS/MS using an electrospray ion source in negative mode.

## Spectral analysis

UV spectra for all polymer sequences were obtained with a Cary 60 spectral analyzer using 100mM sodium bicarbonate ($NaHCO_3$) buffer at pH 9.5 and a 1 x 1 cm quartz cuvette

(concentrations = ~1.2μM). Fluorescence spectra were obtained with a Cary Eclipse Fluorimeter, using the same cuvette and buffer system (concentration = ~25nM).

### Generalized protocol of polyamine coupling to activated NHS fluorophores

In a 200μL microtube was placed a solution of borate buffer (5.3μL, 150mM, pH = 9.0). To this was added a solution of fluorophore NHS ester (4.5 μL, 50mM in DMSO), a solution of the polyamine sequence (1.5 μL, 1.0mM in water) and a solution of magnesium chloride (3.8 μL, 2000mM). The mixture was allowed to incubate at room temperature overnight. The mixture was diluted with water (75 μL) and purified by size exclusion chromatography (Superdex 200 increase 5/150 GL, isocratic elution with PBS, 0.25mL/min, detection at 494nM). Alternative purification can be accomplished by bringing the reaction mixture up in sodium chloride (final concentration, 100mM) and diluting with cold ethanol (9:1 v/v) and allowing to stand at -20˚C for 2h. The polyphosphate precipitate is isolated by centrifugation, the supernatant disposed and the solid pellet reconstituted in water prior to analysis.

### Computational analysis

The conformation of backbone was simulated using a semi empirical molecular orbital program MO-G, which was produced by Fujitsu based on MOPAC2002. For solving the Schrödinger equation, we selected PM6 as Hamiltonian, and adopted COSMO method for solvent (water) effect calculation, considering intramolecular hydrogen bonding. Number of iterations were depended on each of the molecules. Normal calculation continued until the Gradient Normal parameter became 1 or less.

### Conclusion

Presented herein is a novel programmable polymer system which is assembled using automated DNA synthesizers. Manipulation of charge density using a series of simple phosphoramidite building blocks (Fig 3) allows to some extent, control of macromolecular structure. We used this unique characteristic to design polymers to control spacing of pendant fluorophores which prevented contact quenching energy dissipation and allowed for linear enhancement of brightness. This concept was further developed to include incorporation of amino (and other) attachment points at any point in the polymer sequence by simply programming the DNA synthesizer with the desired sequence. It is our belief that this system will have far reaching impact due to the variety of "payloads" that can be loaded onto the prepositioned attachment points. Furthermore, due to development of orthogonal (3) attachment chemistries, three different payloads can be loaded at any given point in the polymer. This enables polymer construction with multiple dye types which has obvious implications for tandem dye (via FRET) applications (manuscript in preparation). We have also loaded cytotoxic agents onto the polyamino constructs and proven that ADC based therapeutics are possible and do indeed specifically target and kill cancer cells. We believe that many fields of research will benefit from this technology platform including materials science, probe based IVD assay development, imaging, and many others.

### Supporting information

**S1 Fig. Table of theoretical and observed mass spec data for compounds 1–11.**
(TIF)

**S2 Fig. Table of theoretical and observed mass spec data for compounds shown in Fig 5A.**
(TIF)

**S3 Fig. Emission max values for GEN 1 and GEN II 10x polymers at increasing MgCl2 concentration.** The presence of divalent MgCl2 had a significant downward effect on the emissions of both GEN I and GEN II polymer systems, but the degree and regularity of the effect was different between the two constructs: Where the emission values for GEN I (presumably the more rigid construct) fell more sharply, they also fell in a more uniform fashion, whereas the emission values for GEN II were less uniform and with a less drastic drop in efficiency. (TIF)

**S4 Fig. Emission max values for HEG based GEN II polymers 3x, 5x, and 10x at 10nM concentration.** (TIF)

**S5 Fig. Reverse phase HPLC showing approximately 60% crude purity for a GENII 10x construct made on a AKTA 100 DNA synthesizer on 13 μmol scale.** (TIF)

**S6 Fig. Emission spectra of equal concentration solutions of GENIII constructs 3x,5x,7x,8x,10x.** (TIF)

**S7 Fig. Table of theoretical and observed mass spec data for compounds in Fig 10.** (TIF)

**S1 Data. Emission spectra data.** (XLSX)

**S1 File. LCMS data.** (DOCX)

## Author Contributions

**Conceptualization:** Tracy Matray, Sharat Singh, Hesham Sherif, Michael VanBrunt.

**Data curation:** Kenneth Farber, Erin Kwang, Michael VanBrunt.

**Formal analysis:** Tracy Matray, Sharat Singh, Hesham Sherif, Kenneth Farber, Erin Kwang, Michael VanBrunt.

**Funding acquisition:** Tracy Matray.

**Investigation:** Tracy Matray, Hesham Sherif, Kenneth Farber, Erin Kwang, Michael VanBrunt, Eriko Matsui.

**Methodology:** Tracy Matray, Sharat Singh, Hesham Sherif, Kenneth Farber, Michael VanBrunt.

**Project administration:** Tracy Matray, Hesham Sherif, Eriko Matsui, Hiroaki Yada.

**Supervision:** Tracy Matray, Hesham Sherif.

**Validation:** Hesham Sherif.

**Writing – original draft:** Tracy Matray.

**Writing – review & editing:** Tracy Matray, Hiroaki Yada.

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
