## [Decision Letter · Decision Letter 0]

23 Sep 2020

PONE-D-20-27312

A New Class of Polymeric Fluorescent Dyes Assembled Using

A DNA Synthesizer

PLOS ONE

Dear Dr. Matray,

Thank you for submitting your manuscript to PLOS ONE. After careful consideration, we feel that it has merit but does not fully meet PLOS ONE’s publication criteria as it currently stands. Therefore, we invite you to submit a revised version of the manuscript that addresses the points raised during the review process.

Ïn particular, the authors should make sure to address the following points:

a) Increasing the resolution and quality of images of all figures has been brought up by both reviewers, so this point should be addressed before acceptance.

b) The synthesis portion should include further characterization details. The authors should also indicate a possible path towards synthesis in larger quantities to cater to the envisioned applications (although it is not necessary that they carry out a scale up experimentally at this time). It may also be considered to refer to the materials obtained as oligomers rather than polymers, if the degree of polymerization is relatively low. 

We look forward to receiving your revised manuscript.

Kind regards,

Marco Bonizzoni, Ph.D.

Academic Editor

PLOS ONE

Journal Requirements:

2. Please amend your Data availability statement to provide details of how others may access the data from this study. We note for instance that no supporting information has been provided.

3.Thank you for providing the following Funding Statement: 

[All work funded by Sony Corporation and Sony Imaging Products & Solutions Inc.  

The funders had no role in study design, data collection and analysis, decision to publish, or preparation of the manuscript.].

We note that one or more of the authors is affiliated with the funding organization, indicating the funder may have had some role in the design, data collection, analysis or preparation of your manuscript for publication; in other words, the funder played an indirect role through the participation of the co-authors.

If the funding organization did not play a role in the study design, data collection and analysis, decision to publish, or preparation of the manuscript and only provided financial support in the form of authors' salaries and/or research materials, please review your statements relating to the author contributions, and ensure you have specifically and accurately indicated the role(s) that these authors had in your study in the Author Contributions section of the online submission form. Please make any necessary amendments directly within this section of the online submission form.  Please also update your Funding Statement to include the following statement: “The funder provided support in the form of salaries for authors [insert relevant initials], but did not have any additional role in the study design, data collection and analysis, decision to publish, or preparation of the manuscript. The specific roles of these authors are articulated in the ‘author contributions’ section.”

If the funding organization did have an additional role, please state and explain that role within your Funding Statement.

Please also provide an updated Competing Interests Statement declaring this commercial affiliation along with any other relevant declarations relating to employment, consultancy, patents, products in development, or marketed products, etc.  

Reviewers' comments:

Reviewer's Responses to Questions

**Comments to the Author**

1. Is the manuscript technically sound, and do the data support the conclusions?

Reviewer #1: Partly

Reviewer #2: Yes

2. Has the statistical analysis been performed appropriately and rigorously? 

Reviewer #1: N/A

Reviewer #2: N/A

3. Have the authors made all data underlying the findings in their manuscript fully available?

Reviewer #1: No

Reviewer #2: Yes

4. Is the manuscript presented in an intelligible fashion and written in standard English?

Reviewer #1: Yes

Reviewer #2: Yes

5. Review Comments to the Author

Reviewer #1: This paper demonstrates the applicability of a DNA synthesizer for the synthesis of fluorescently-labeled phosphodiester oligomers. The authors successfully prepared a series of oligomers covalently tagged by two fluorescent dyes at the both ends, and discussed the effects of oligomer length and type on quenching of the dyes. This paper reports new oligomers, and thus the authors should provide a summary table of LC-MS results, including expected and experimental molecular weights, purity, yield, for all of the oligomers. They should also provide characterization data for the post-functionalization of polyamines with fluorophores, possibly obtained using LC-MS (and NMR). In addition, the following points should be considered for publication of this work in PLOS ONE.

1. The title should be changed so that it represents the achievements of this work, which I think is the application of a DNA synthesizer for synthesis of various oligomers. This paper does not report any new class of fluorescent dyes, as the fluorescence of the oligomers originates from conventional fluorescent dyes, but not from the oligomer chains.

2. The authors are strongly encouraged to quantitatively analyze the relationship between the fluorophore distance, which may be estimated computationally, and quenching efficiency. The quenching efficiency is well-known to be controlled by distance between the fluorophores and also their mutual orientation. This analysis should help the authors discuss the roles of linker properties in quenching more clearly.

3. In Introduction, the authors should discuss similar quenching studies using other oligomeric linkers like oligonucleotides, peptides and synthetic polymers to highlight the novelty of this work based on phosphodiester oligomers. For protein labeling (Line 50), the authors should also mention genetically encoded fluorescent proteins and other materials such as conducting polymer dots.

4. The authors should consider a possibility of the complexation of PEG with Na+ for the modeling in Figure 7.

5. Improve the quality of the figures. The resolution of many graphs should be higher than 300 dpi. The labels on the x- and/or y-axes are missing for Figures 1A, 4BC and 9. The style of the graphs in Figure 9 should be the same, and Figure 9D should be replaced as the fluorescence intensity is saturated for 10X. Spectra should be given in Figure SM1 instead of the plots. All the captions should provide experimental conditions such as excitation wavelength, concentrations, solution conditions and HEG/PEG (for Figure 9).

Finally, the authors are strongly encouraged to edit the paper more carefully. There are a number of typos (for example, “phosphoramdite” on Line 99) and grammatical issues (for example, “correlate” on Line 198 and “total” on Line 214). Define Gen I, II and III clearly. The synthesized materials sound oligomers, rather than polymers, considering their relatively small polymerization degrees.

Reviewer #2: The manuscript entitled 'A new class of polymeric fluorescent dyes assembled using

5 a DNA synthesizer' submitted by Yada et al describes a novel and facile synthetic method to obtain multivalent (polymeric) fluorescent dye with optimizable properties. Overall, it is a well-written manuscript and sufficient amount of experiments (along controls) have been done. The paper is acceptable for PLOS One with minor revision. These revisions will be needed, in my opinion, in following two areas:

a) Increasing the resolution and quality of images of all figures, particularly fluorescent spectroscopic plots. Apparently, the legends are unreadable, and in others axis-associated fonts are too small.

b) The synthesis is applicable, in its current format, in microscale. A paragraph regarding scalability of the process is requested.

c) Since, the manuscript is aimed for PLOS One, the audience of which has diversified background, a more understandable reaction scheme will improve the impact of the manuscript.

6. PLOS authors have the option to publish the peer review history of their article (what does this mean?). If published, this will include your full peer review and any attached files.

Reviewer #1: No

Reviewer #2: No

---

## [Author Response · Author response to Decision Letter 0]

12 Nov 2020

The authors would like to thank both reviewers and the editor for their time and thoughtful suggestions on making improvements to the manuscript. We have done our best to address most of the points raised in the revised manuscript. Specifically addressing each point raised:

Editor:

a) All figures (and supplementary information) redone to increase resolution and clarity except for the 3 structural in silico simulation figures (they would have to be completely re-simulated and computing time in Japan would be difficult to obtain given the time frame). Also, we believe the images convey the overall macromolecular structure well in our opinion). 

b) Added new Supplementary Information (SM1,SM2,SM7,SM8 and SM9) = mass spec data for oligomers and polymers in Figures (2,5 and 10) also included are purity levels of most of the constructs (SM9). Changed some “polymer” references to “oligomer” for the shorter constructs. Added statement on scalability of these types of compounds (line 217-218).

Journal Requirements:

1. All updated figures and SMs were passed through PACE

2. Data availability statement amended as requested. Raw data in the form of excel spread sheets is provided for all emission curves in Figures 5,6 and 10 (SM8). LCMS data for the constructs is included as well.

3. Modified the Funding statement to include:

“The funder provided support in the form of salaries for all authors, but did not have any additional role in the study design, data collection and analysis, decision to publish, or preparation of the manuscript. The specific roles of these authors are articulated in the ‘author contributions’ section.”

Also modified author contributions as requested.

4. Modified the Competing Interests Statement to include:

"The commercial affiliation does not alter our adherence to PLOS ONE policies on sharing data and materials.”

Reviewer 1:

1. All figures (and supplementary materials) redone to increase resolution and clarity except for the 3 structural in silico simulation figures (they would have to be completely re-simulated and computing time in Japan would be difficult to obtain given the time frame). Also, we believe the images convey the overall macromolecular structure well in our opinion). Per reviewers and editor requests

2. Added new Supplementary Information (SM1,SM2,SM7 and SM9) = mass spec data and HPLC chromatograms for oligomers and polymers in Figures (2,5 and 10) (yield data not possible for microscale synthesis but is provided for larger scale synthesis along with purity data in SM5. Purity data also included in SM9.

3. Modified title and changed some “polymer” references to “oligomer” for the shorter constructs. Many of the constructs are over 50 residues long which we left as polymer 

4. Added additional x axis and y axis legends to Figures. 

5. The suggestion made to try and qualitatively analyze the quenching efficiency based on distance calculations while valid, would be difficult due to the computation time required as mentioned above. We feel the images and emission data in the figures (2 and 6) are sufficient to give readers an understanding of the linear control of quenching in this system. 

6. Added mention of protein based fluorescent compounds and potential role of sodium ions in Figure 8. 

7. Minor edits as pointed out by reviewers for added clarity 

Reviewer 2:

a. All figures (and supplementary materials) redone to increase resolution and clarity except for the 3 structural in silico simulation figures (they would have to be completely re-simulated and computing time in Japan would be difficult to obtain given the time frame). Also, we believe the images convey the overall macromolecular structure well in our opinion). Per reviewers and editor requests

b. Added statement on scalability of these types of compounds (line 217-218)

c. Added an additional Figure (1) – coupling chemistry scheme - 

d. Added brief text description of chemical steps – lines 58-66

e. Added new reference (7) Caruthers paper for original published report of phosphoramidite chemistry

---

## [Editor Report · Decision Letter 1]

18 Nov 2020

A Novel Class of Polymeric Fluorescent Dyes Assembled Using

A DNA Synthesizer

PONE-D-20-27312R1

Dear Dr. Matray,

We’re pleased to inform you that your manuscript has been judged scientifically suitable for publication and will be formally accepted for publication once it meets all outstanding technical requirements.

Kind regards,

Marco Bonizzoni, Ph.D.

Academic Editor

PLOS ONE

---

## [Editor Report · Acceptance letter]

23 Nov 2020

PONE-D-20-27312R1 

A novel class of polymeric fluorescent dyes assembled usinga DNA synthesizer

Dear Dr. Matray:

I'm pleased to inform you that your manuscript has been deemed suitable for publication in PLOS ONE. Congratulations! Your manuscript is now with our production department. 

Kind regards, 

on behalf of

Dr. Marco Bonizzoni 

Academic Editor

PLOS ONE